# Titanium Substratum Roughness as a Determinant of Human Gingival Fibroblast Fibronectin and α-Smooth Muscle Actin Expression

**DOI:** 10.3390/ma14216447

**Published:** 2021-10-27

**Authors:** Hong Li, Chengyu Guo, Yuchen Zhou, Hao Sun, Robin Hong, Douglas William Hamilton

**Affiliations:** 1Hangzhou Dental Hospital Xiaoshan Branch, School of Stomatology, Zhejiang Chinese Medical University, Hangzhou 310000, China; lihong8689@163.com; 2Schulich Dentistry, Schulich School of Medicine and Dentistry, University of Western Ontario, London, ON N6A 5C1, Canada; cguo83@uwo.ca (C.G.); yzhou763@uwo.ca (Y.Z.); hsun297@uwo.ca (H.S.); rhong1718@gmail.com (R.H.); 3Department of Anatomy & Cell Biology, Schulich School of Medicine and Dentistry, University of Western Ontario, London, ON N6A 5C1, Canada

**Keywords:** titanium surface roughness, gingiva, fibrosis, adhesion stability, myofibroblasts

## Abstract

The most appropriate surface treatment to enhance gingival connective tissue formation on the abutment of dental implants remains undefined, with healing associated with a scar-like response. We have previously shown that topographies with an arithmetic average of the absolute profile height deviations (R_a_) = 4.0 induces an anti-fibrotic phenotype in human gingival fibroblasts (HGFs) by causing nascent adhesion formation. With bacterial colonization considerations, we hypothesized that a lower R_a_ could be identified that would alter adhesion stability and promote a matrix remodeling phenotype. Focal adhesions (FAs) area decreased with increasing roughness, although no differences in cell attachment or proliferation were observed. Alpha smooth muscle actin (α-SMA) protein levels were significantly reduced on R_a_ = 3.0 and 4.0 vs. 0.1 (*p* < 0.05), with incorporation of α-SMA into stress fibers most prominent on R_a_ = 0.1. Fibronectin protein levels were reduced on 3.0 and 4.0 vs. 0.1 (*p* < 0.05), and R_a_ = 1.5 and deeper significantly altered fibronectin deposition. Addition of exogenous TGF-β3 increased HGF adhesion size on 0.1 surfaces, but not on any other topography. We conclude that R_a_ = 1.5 is sufficient to reduce adhesion size and inhibit α-SMA incorporation into stress fibers in HGFs, but 3.0 is required in the presence of exogenous TGF-β3. Our findings have implications for inhibiting fibrotic tissue formation surrounding percutaneous devices such as dental implants.

## 1. Introduction

A robust attachment of gingival connective tissue to the transmucosal region of a dental implant plays an essential role in maximizing the longevity of the device [1]. Without stable attachment of connective tissue to the abutment surface, downgrowth of the overlying oral epithelium can occur [2], which significantly erodes the barrier to bacterial colonization, thereby increasing inflammation and the development of peri-implantitis [3,4]. Compared with research on optimizing the surface characteristics of the intraosseous component to maximize bone formation [5], fewer investigations have focused on the transmucosal abutment and on enhancing the attachment of the gingival connective tissue [6]. Surrounding both machined and polished abutments, the adjacent connective tissue has a composition similar to scar tissue [7,8,9].

In tissues such as gingiva, mesenchymal cells and specifically the resident fibroblast populations are the central cell types that mediate tissue repair and fibrosis [10,11]. Post implantation, gingival fibroblasts attach to the abutment surface of the implant and establish an extracellular matrix (ECM) in direct apposition to the implant surface [12,13,14]. It is known that substratum topography directly influences fibroblast phenotype [15], predominantly through alterations in adhesion formation and downstream signaling [16]. Altered adhesion stability and composition are also known to be critical in the transition of fibroblasts to an ECM secreting, contractile α-smooth muscle actin (α-SMA) expressing myofibroblast [17] phenotype, with development of supermature adhesions associated with this phenotype. Additionally, an essential contributor to gingival healing [9], transforming growth factor beta (TGF-β) signaling induces transition of fibroblasts to myofibroblasts [18,19]. TGF-β selectively promotes fibroblast to myofibroblast transition through both SMAD phosphorylation (canonical) and adhesive signaling; a non-canonical pathway involving modification of adhesion formation, integrin subunit engagement and focal adhesion kinase phosphorylation (FAK) [20]. Therefore, topography of the implant surface and TGF-β signaling could be important determinants of fibroblast phenotype and connective tissue healing at the abutment of the implant through HGF adhesion modification.

We have previously shown that sand blast, large grit, acid-etch (SLA) roughened titanium topographies can be utilized to reduce myofibroblast differentiation in HGFs. The roughened topographical features induce nascent adhesion formation in HGFs which attenuates α-SMA expression, fibronectin synthesis, stress fiber assembly, and is concomitant with an upregulation of genes associated with matrix remodeling [21]. This study provides direct evidence that alterations in titanium substratum roughness can be utilized to reduce myofibroblast differentiation of HGFs.

While application of SLA to the abutment of the transmucosal region of a dental implant is biologically relevant to reduce myofibroblast differentiation and scar tissue formation, it provides a large surface area that could increase leading to peri-implantitis. Titanium, with lower arithmetic average of the absolute profile height deviations (Ra) = 3.39 μm (Tis-OPAAE) while shown to promote HGF adhesion, also permitted bacterial colonization [22]. Our previous observations suggest that substratum roughness restricts sites for HGFs to form adhesions inhibiting myofibroblast differentiation as opposed to the average topographical depth of the features [21]. We hypothesized that roughened topographical features could be produced with a lower R_a_ that would inhibit a scarring phenotype in HGF, while reducing the area for bacterial colonization. Using α-SMA and fibronectin as a readout, we investigated the influence of varying the R_a_ of titanium on HGF adhesion formation and downstream phenotype.

## 2. Materials and Methods

### 2.1. Preparation of Titanium Surfaces

Fabrication of each topography was performed as previously described [16,21]. Briefly, Commercially-pure titanium (Cp-Ti) (Baoji Titanium Industry Co., Ltd., Baoji, China) was cut into 1-mm thick discs from Cp-Ti rods 15 mm in diameter. Cut discs were then sand blasted with 1 of 3 different sizes of Al_2_O_3_ particles (45 μm, 125 μm, and 250 μm) under 0.7 MPa for 2 min (Qinggong Machinery, Qingdao, China) and cleaned in an ultrasonic bath for 15 min. Each group contained 20 samples. Each sample was then acid-etched for 20 min at 100 °C in a 1:1 mixture of 60% sulfuric acid and 10% hydrochloric acid. Post-processing, the roughnesses of the titanium samples were detected by a TR200 Portable TIME3200 Non-Destructive Roughness Tester (TIME High Technology Ltd., Beijing, China). This process resulted in topographies with R_a_ = 0.5, 1.5 and 3.0. As controls, pickled titanium (PT, R_a_ = 0.1) and SLA topographies (R_a_ = 4.0) were kindly supplied by Institut Straumman AG (Basel, Switzerland). Scanning electron microscopy was performed as previously described [23]. The samples were viewed using a Hitachi 3400-N scanning electron microscope at 4 kV accelerating voltage. The topographic features of the PT and SLA surfaces are shown in Figure 1A–E and the EDX of Cp titanium are shown in Figure 1F. The characteristics of the fabricated topographies has been previously published.

### 2.2. Human Gingival Fibroblast Isolation

Clinically healthy gingiva (*n* = 6) was obtained with informed consent from six patients undergoing periodontal or implant therapies at the Oral Surgery Clinic at the University of Western Ontario. The use of all tissue material was in accordance with the guidelines of the University’s Research Ethics Board for Health Sciences Research involving Human Subjects requiring informed patient consent. Human gingival fibroblasts (HGFs) were extracted from the tissue using explant cultures as previously described [16,21]. HGFs were maintained in high glucose Dulbecco’s modified Eagle’s medium (DMEM) (Invitrogen, Carlsbad, CA, USA) supplemented with 10% fetal bovine serum and 1% antibiotic/antimycotic (100 μg/mL penicillin, and 100 μg/mL streptomycin, and 0.25 mg/mL amphotericin B), at 37 °C in a humidified atmosphere of 95% air 5% CO_2_. Cells were removed from the growth surface with 0.25% trypsin (Gibco) and 0.1% glucose in citrate-saline buffer (pH 7.8). HGFs were used between passages 2–7 for all experiments. In defined experiments, HGFs were left to attach overnight before treatment with 5 ng/mL recombinant human TGFβ1 or TGFβ3 (R&D Systems, Minneapolis, MN, USA).

### 2.3. Cell Attachment and Proliferation

HGFs attachment and growth was assessed on all surfaces using CyQUANT^®^ Cell Proliferation Assay Kit (C7026; Molecular Probes, Eugene, OR, USA) according to the manufacturer’s instructions and our previous experience [24]. Briefly, in parallel cultures, 10,000 (Attachment assays) or 5000 (Proliferation studies) HGFs was seeded onto each disc and at each designated timepoint, media was removed gently with a pipette gun, followed rinse by cold PBS twice to remove unattached cells. A total of 5000 cells were seeded for proliferation to have significant surfaces area for growth as HGFs are contact inhibited. Lysates were then transferred into a new 24-well plate and stored at −80 °C. DNA content was determined by performing CyQUANT^®^ Cell Proliferation Assay Kit (C7026; Molecular Probes, Eugene, OR, USA). Cell numbers were extrapolated using a standard curve. Three independent experiments were performed, with 6 replicates per condition per experiment. For attachment, timepoints selected were 1, 6 and 24 h, and for cell proliferation, 1, 3 and 7 days.

### 2.4. Immunocytochemistry

HGFs were seeded on each topography at 10,000 cells for 24 h and 1 week. At the end of the designated timepoints, samples were fixed with 4% paraformaldehyde, permeabilized with 0.1% Triton X-100 and blocked with 1% bovine serum albumin. Fixed and permeabilized cells were labeled with antibodies against fibronectin (sc-8422; Santa Cruz Biotechnology, Dallas, TX, USA; 1:1000), α-SMA (A5228; Sigma-Aldrich, Oakville, ON, Canada; 1:1000), vinculin (V4505; Sigma-Aldrich; 1:100, Oakville, ON, Canada), and phospho-SMAD3 (phospho Ser423/Ser425) (ab52903; Abcam 1:200). These signals were detected using primary antibodies followed by species appropriate IgG conjugated to appropriate Cy5, FITC or TRITC secondary antibodies (Molecular Probes; 1:200 dilution). Images were taken on a Carl Zeiss Observer Z1 microscope using AxioVision Relative software (version 4.8, Toronto, ON, Canada).

### 2.5. Western Blotting

HGFs were washed twice with 4 °C PBS and protein isolated in a RIPA buffer (Sigma-Aldrich, Oakville, ON, Canada) containing protease (Roche Diagnostics GmbH, Mannheim, Germany) and phosphatase inhibitor (Calbiocam, Billerica, MA, USA) cocktails. Protein concentration was determined by Pierce^®^ BCA Protein assay kit (Pierce, Waltham, MA, USA). An amount of 25 μg proteins of each sample were separated by sodium dodecyl sulfate polyacrylamide gel electrophoresis (SDS-PAGE) and transferred to nitrocellulose membranes. The membranes were washed with Tris-buffered saline containing 0.05% Tween-20 (TBS-T) and blocked with 5% dried milk in TBS-T. Primary antibodies for fibronectin (sc-8422; Santa Cruz Biotechnology; 1:1000, Dallas, TX, USA), α-SMA (A5228; Sigma-Aldrich, 1:1000, Oakville, ON, Canada) and GAPDH (MAB374; Millipore; 1:2000, Toronto, ON, Canada) were used. Detection was performed with appropriate perioxidase-conjugated secondary antibodies (Jackson Immuno Research, West Grove, PA, USA; 1:2000), which were developed with SuperSignal Western Pico Chemiluminescence Substrate (Pierce, Waltham, MA, USA).

### 2.6. Adhesion Size Quantification

To calculate adhesion size on each topography, cells were fixed in 4% paraformaldehyde and labeled with antibodies specific to vinculin as described in Section 2.4. Planar areas of FAs were calculated using ImageJ software (National Institutes of Health, Bethesda, MD, USA). In brief, vinculin plaques were thresholded and converted to binary images, with size measured using standard ImageJ macros.

### 2.7. Statistical Analysis

After analysis of data distribution using Kurtosis, statistical analysis was performed by one-way ANOVA, as appropriate, followed by a Bonferroni correction, using Graphpad Software v.5 (Graphpad Software, San Diego, CA, USA) (*p* ≤ 0.05 was considered significant). Data is expressed as the mean ± standard deviation of three independent experiments with cells isolated from three different patients. Individual experiments included at least three replicates per condition.

## 3. Results

### 3.1. Increasing R_a_ of Titanium Does Not Influence HGF Attachment or the Temporal Increase in Cell Number

Attachment of HGFs was assessed at 1-, 6- and 24-h post seeding. HGFs adhered to all surfaces within 24 h, but no differences were observed between any tested substratum R_a_ (Figure 2A). Assessment of temporal increase in cell number similarly showed no differences in HGF growth between surfaces (Figure 2B).

### 3.2. Adhesion Formation

The size of vinculin-containing FAs through which cells attach to each surface was assessed. Immunocytochemistry with specific antibodies to vinculin showed that larger and more stable adhesions were formed on R_a_ = 0.1 than was evident on any other level of roughness (Figure 3). With increasing R_a_ from 0.1, 0.5, 1.5, 3.0, to 4.0, vinculin-associated plaques became smaller in size. On R_a_ = 0.1 HGFs were well spread and in close contact with the surface, but with increasing depth, cells formed adhesion predominantly at the cell periphery in lamellapodia only and were apparently not in close contact with the underlying substratum in the middle of the cells (Figure 3).

### 3.3. Phospho-SMAD3 Phosphorylation Translocates to the Nucleus in HGFs on All Levels of Substratum Roughness

As TGF-β signaling is an important determinant of wound healing, we next assessed whether substratum roughness had any influence on activation of phosphor-SMAD3. On all tested topographies, immunocytochemistry demonstrated nuclear translocation of phospho-SMAD3, which occurs upstream of matrix synthesis and α-SMA activation, indicating activation of canonical TGF-β1 signaling irrespective of substratum roughness (Figure 4).

### 3.4. α-SMA Protein Levels Decrease with Increasing Substratum Roughness

Transition of HGFs to a contractile α-SMA myofibroblast phenotype is associated with scarring, a tissue structure known to exist in apposition to implant abutments [7]. The effect of substratum roughness on α-SMA levels was assessed at 1-day and 1-week post seeding. At 1-day post seeding, α-SMA protein levels were significantly reduced only on surfaces of R_a_ = 4.0 in comparison with 0.1 (*p* < 0.05) (Figure 5A,C). At 1-week post seeding, α-SMA levels were significantly reduced on surfaces on R_a_ = 3.0 and 4.0 versus 0.1 (*p* < 0.05) (Figure 5B,C). Upregulation of α-SMA results in a proto-myofibroblast phenotype [25], unless α-SMA is incorporated into stress fibers. Assessment of α-SMA incorporation into stress fibers was performed using immunocytochemistry (ICC) of cells at 1-week post seeding. As we have previously shown [21], on titanium with R_a_ = 0.1, numerous cells incorporate α-SMA into stress fibers assuming a full myofibroblast phenotype (Figure 5D). However, on R_a_ = 1.5 and rougher, no incorporation of α-SMA into stress fibers was observed, demonstrating that the cells exhibit a proto-myofibroblast phenotype.

### 3.5. Fibronectin Protein Deposition Is Altered by Increasing Substratum Roughness

A hallmark of myofibroblasts is an increase in deposition of ECM and we have previously shown that fibronectin protein levels are an appropriate readout of this phenotype within 24-h post seeding [21]. Assessment of fibronectin levels was performed at 1-day and 1-week post seeding using Western blotting and ICC. Western blot analysis of fibronectin levels in total lysates (cells + ECM secreted fibronectin) showed that fibronectin protein levels were similar on all roughnesses tested except 3.0 and 4.0 (*p* < 0.05) versus 0.1 controls (Figure 6A,C). At 1-week post seeding, a significant reduction in fibronectin protein levels was observed only on 4.0 versus 0.1 (*p* < 0.05) (Figure 6B,C). ICC with antibodies specific for extracellular fibronectin demonstrated that fibronectin fibrils were more developed in cells cultured on 0.1 at 24 h, with a concomitant decrease in fibrils with increasing roughness (Figure 6D). At 1-week post seeding, on R_a_ = 0.1 HGFs exhibited fibronectin protein deposition, which aligned predominantly in the direction of the cell long axis. On surfaces of R_a_ = 1.5 and deeper, fibronectin organization became more disorganized and linear alignment of the fibronectin fibrils was attenuated.

### 3.6. Substratum Roughness and TGF-β-Signaling, Combine to Regulate α-SMA Incorporation into Stressfibres in HGFs

It is known that cells increase adhesion size in response to TGF-β [26], so we next assessed whether substratum roughness or TGF-β was determining adhesion size. We next assessed the influence of two TGF-β isoforms on vinculin-containing plaques on all topographies (Figure 7). Addition of exogenous TGF-β3, but not TGF-β1, significantly increased the adhesion planar area on R_a_ = 0.1 compared to untreated HGFs (Figure 7A,B). On surfaces of R_a_ = 0.5, 1.5, 3.0 and 4.0, the addition of TGF-β1 had no effect on adhesion size. The addition of exogenous TGF-β1 and TGF-β3 to cells cultured on 0.1 resulted in increased fibronectin deposition and stress fiber formation over controls (Figure 8). On surfaces of R_a_ = 1.5, there was reduced α-SMA incorporation into stress fibers which was increased with the addition of TGF-β1, but not TGF-β3 (Figure 8). When cultured on surfaces of R_a_ = 3.0, no evidence of α-SMA incorporation into stress fibers was seen, even in the presence of either TGF-β isoform (Figure 8).

## 4. Discussion

One of the main factors governing the longevity of dental implants is formation and maintenance of a healthy soft tissue/implant interface. Fibroblasts attachment to titanium is a critical determinant of their phenotype, and the topography of the abutment will influence HGF contractility, ECM synthesis and response to TGF-β stimulation, processes essential for connective tissue formation. The focus of this study was to investigate how variations in titanium substratum roughness alters gingival fibroblast adhesion and the regulation of fibronectin and α-SMA expression in HGFs with a long-term focus on treatments for enhancing gingival connective tissue formation on implant abutments.

Based on their association with tissue healing, myofibroblast differentiation and fibrosis, α-SMA and fibronectin were selected as the primary phenotypic measures. Alterations in adhesion size and composition are required for both myofibroblast differentiation and extracellular matrix synthesis [11,17,20,27], making optimization of substratum topography a viable method with which to attenuate adhesion and HGF contractility [21,28,29,30]. Understanding mechanistically how titanium of different R_a_ levels influences HGF adhesion stability and maturation is essential for optimizing gingival tissue formation, although the most appropriate R_a_ has not been specifically determined in relation to adhesion and myofibroblast activity of HGFs [6].

The size of a FA is an indication of its maturity and stability, although a balance in HGF adhesion formation needs to be achieved as supermature adhesions are associated with increased cell contractility and a pro-fibrotic response that results in scar formation [19,20]. We demonstrate here that HGFs on R_a_ = 0.1 formed significantly larger FAs compared to cells on R_a_ = 1.5 and deeper (Figure 7b), a similar finding reported by other groups using varying types of roughened titanium [21,31]. We have previously shown that on R_a_ = 0.1 adhesions are more stable, with vinculin linked adhesion sites associated with co-localization of the nucleation promoting factor phospho-cortactin [32], a cytosolic protein implicated in stabilization of the F-actin cytoskeleton [33]. While roughening the titanium increases the overall surface area, it effectively reduces site availability for HGF adhesion, likely due to peaks produced by sand blasting and acid-etching. Such topographies limit maturation of the adhesion sites and in gingival fibroblasts attenuates phospho-cortactin co-localization with FAs resulting in reduced adhesion stability [21]. HGFs on SLA (R_a_ = 4.0) form small unstable vinculin-containing FAs, resulting in changes in intracellular signaling upstream of ECM remodeling [16,21]. We suggest that substratum topographies of a minimum R_a_ = 1.5 are required to prevent supermature adhesion formation in HGFs.

Fibroblast to myofibroblast transition is associated with TGF-β signaling through two distinct but interconnected pathways with respect to α-SMA upregulation and its incorporation into stress fibers: canonical signaling through the transcription factor phospho-SMAD3 [34] and adhesive signaling via integrin engagement and FAK phosphorylation [35,36]. While TGF-β stimulation through canonical SMAD signaling upregulates α-SMA, it requires adhesive based signaling for the protein to be incorporated into stress fiber bundles. Interestingly phospho-SMAD3 was present in the nucleus of cells on all tested surfaces although, α-SMA positive stress fibers were only present in HGFs cultured on R_a_ = 0.1 surfaces, suggesting that even low levels of roughness (R_a_ = 0.5) are potentially sufficient to disrupt adhesion formation that permits HGF increased intracellular tension, supermature adhesion formation and α-SMA incorporation into stress fibers. This is consistent with in vivo performance of smoother surfaces which do not promote strong connective tissue attachment to abutments [12,14]. However, in the presence of exogenous TGF-β1 and β3, it is only on R_a_ = 3.0 and deeper where this reduction in stress fiber formation becomes dominant, highlighting the complexity in the interaction between TGF-β, adhesion formation, intracellular signaling and myofibroblast differentiation in HGFs. While in theory, R_a_ = 1.5 would be sufficient to reduce adhesion size, this R_a_ does not represent an absolute barrier to myofibroblast differentiation, particularly if higher levels of TGF-β1 and β3 are present.

The exact role of TGF-β isoforms in gingival healing remains to be fully elucidated, particularly with regards to healing of gingiva around implant abutments. Three specific TGF-β isoforms have been identified, with specific differences in their role in wound healing particularly in skin [37]. Indeed, much work on TGF-β isoforms in gingival tissue relates to drug-induced gingival enlargement, which is associated with increased ECM production [38], but α-SMA upregulation [39,40]. TGF-β1 is considered to be pro-fibrotic and some evidence suggests that TGF-β3 is anti-fibrotic, although the latter remains somewhat controversial with conflicting results in the literature [41,42,43]. With increased adhesion size as a readout, we show here that TGF-β3, but not TGF-β1, increase vinculin-containing FAs in HGFs and only on surfaces of R_a_ = 0.1. However, HGFs on R_a_ = 0.1 increased incorporation of α-SMA into stress fibers, which for TGF-β1 was independent of increased adhesion size. Both isoforms of TGF-β also increased fibronectin synthesis over a 7-day time period in comparison with untreated controls, although the effects of this difference were diminished on R_a_ = 1.5 and 3.0. In comparison with dermal wounds, there is an increase in the ratio of TGF-β3 to TGF-β1 in gingival healing [44], suggesting that the predominant active isoform in gingival healing is TGF-β3, which would align with our observations of increased adhesion size in HGFs when stimulated with TGF-β3, as well as increased fibronectin deposition. Other studies have shown that TGF-β3 is upregulated in gingival healing in red Duroc pigs [45], but whether TGF-β3 results in reduced scarring in comparison with TGF-β1 remains controversial. As gingival tissue is associated with reduced scarring compared to skin [46], increased activity of TGF-β3 could be part of the molecular pathway associated with this observation.

Binding to fibronectin is also required for induction of α-SMA [26]. Here, we show a higher level of cell-secreted fibronectin by HGFs on R_a_ = 0.1 compared to surfaces of R_a_ = 3.0 and 4.0, particularly at 24-h post seeding. While HGFs do form a fibronectin matrix on all surfaces at 7 days, there is an initial lag phase evident in fibrillogenesis at 24 h when cells are cultured on any level of substratum roughness. Fibronectin has been shown to be deposited directly onto roughened titanium implant surfaces at 4-days post implantation in canine models, with well-formed collagen evident only at the abutment after 14 days [47]. How substratum topography influences fibronectin dynamics is controversial, with studies showing that double acid-etched rough surfaces induce a greater collagen type I and fibronectin production [30]. In contrast, we and others have shown that increased roughness reduces fibronectin deposition compared with smooth surfaces, as well as resulting in more disorganized fibril orientation [21,48]. Interestingly, in the study by Ramaglia et al., they compared their double acid-etched surfaces to machined surfaces [30] which still possess significant topography. Prior studies by Chou and colleagues demonstrated that, on titanium surfaces, the stability of fibronectin mRNA is dependent on the substratum topography on which the cells are cultured [49,50]. It is clear that understanding how changes in substratum topography influence fibronectin production and ECM synthesis in general requires further investigation to optimize abutment integration.

## 5. Conclusions

Our study provides evidence that reducing the size of adhesions that HGFs that can form on titanium can be used to reduce pro-fibrotic responses around implants. This must be considered a balance as complete reduction of a fibrotic response to implant surfaces could be deleterious for gingival healing, increasing the likelihood of early implant failure due to bacterial colonization below the gingiva. However, based on our results in this study, only the application of R_a_ = 3.0 or deeper would represent a topography for preventing α-SMA mediated contraction while still promoting fibronectin synthesis. With regards to potential utility of these topographies, future studies will assess whether these surfaces result in significant bacterial colonization as well as how these surfaces function in vivo in a relevant pre-clinical animal model. In conclusion, by limiting the surface area available for adhesion formation, fibrotic phenotype can be modulated in HGFs cultured on titanium, which has potential relevance for all percutaneous devices.

## Figures and Tables

**Figure 1 materials-14-06447-f001:**
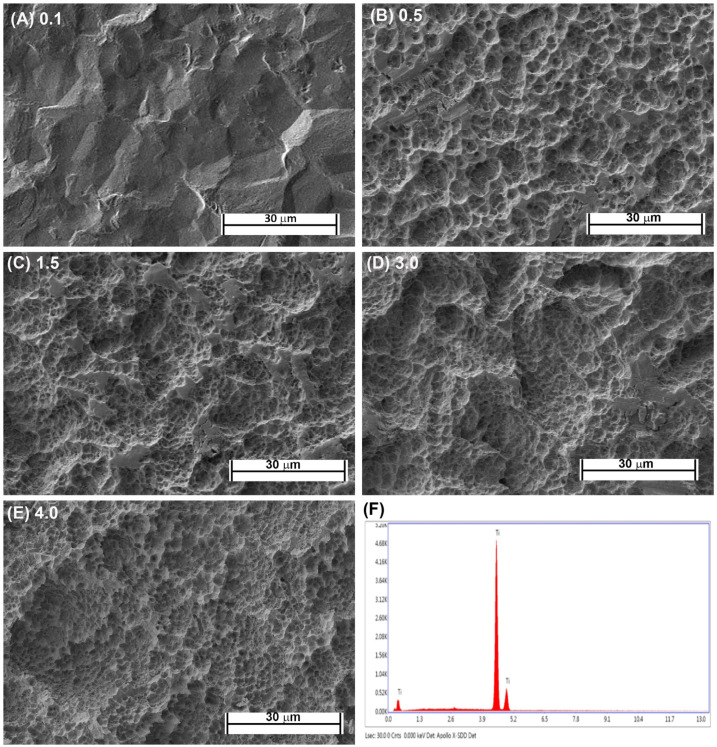
Characterization of titanium topographies used in the study. Scanning electron microscope images on (**A**) 0.1 (PT, Institut Straumann AG), (**B**) 0.5, (**C**) 1.5, (**D**), 3.0 and (**E**) 4.0 (SLA, Institut Straumann AG). In (**F**), XPS spectra is shown confirming surfaces are titanium oxide.

**Figure 2 materials-14-06447-f002:**
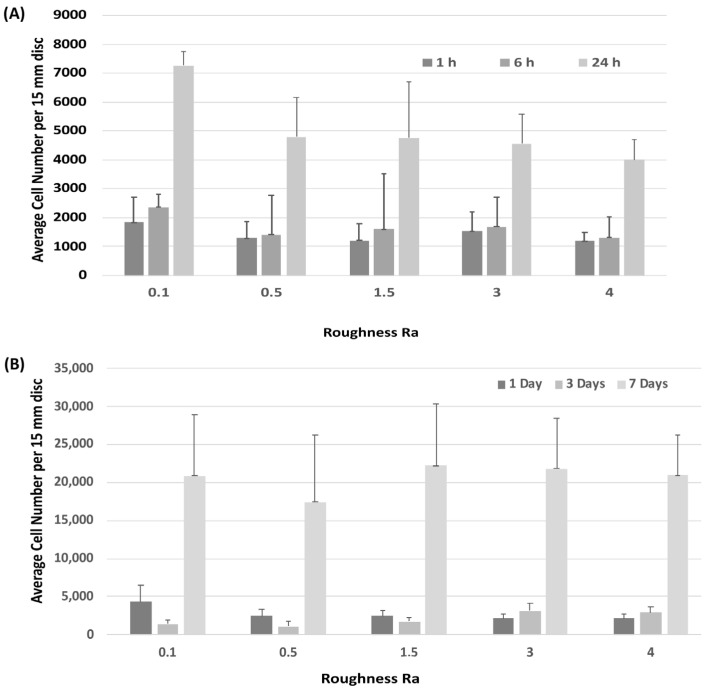
Influence of titanium substratum roughness on cell adhesion and temporal increase in cell number. Cells were seeded on each type of topography for (**A**) 1, 6 and 24 h to assess attachment, and (**B**) 1, 3 and 7 days to assess increase in cell number. No differences in initial attachment or cell proliferation of HGFs on the topographies were quantified. One-way ANOVA followed by a Bonferroni adjustment.

**Figure 3 materials-14-06447-f003:**
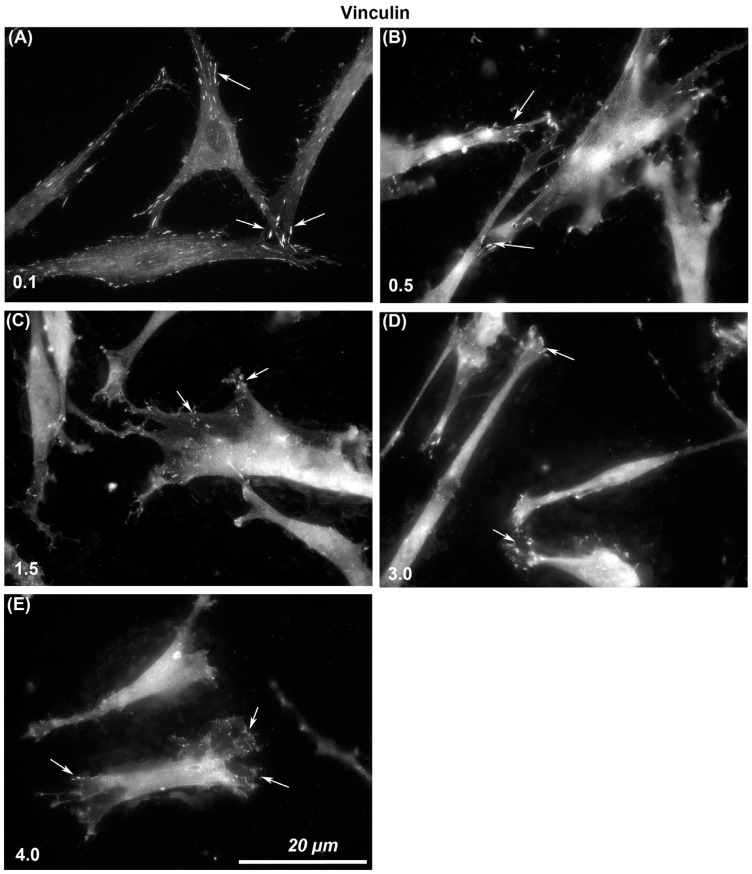
Influence of titanium substratum roughness on adhesion formation in HGFs. Cells were seeded on surfaces with an R_a_ of (**A**) 0.1, (**B**) 0.5, (**C**) 1.5, (**D**) 3.0 (**E**) 4.0, and cultured for 24 h before being labeled with antibodies specific for vinculin. Size of adhesion declined with increasing substratum roughness. White arrows examples of adhesion formation.

**Figure 4 materials-14-06447-f004:**
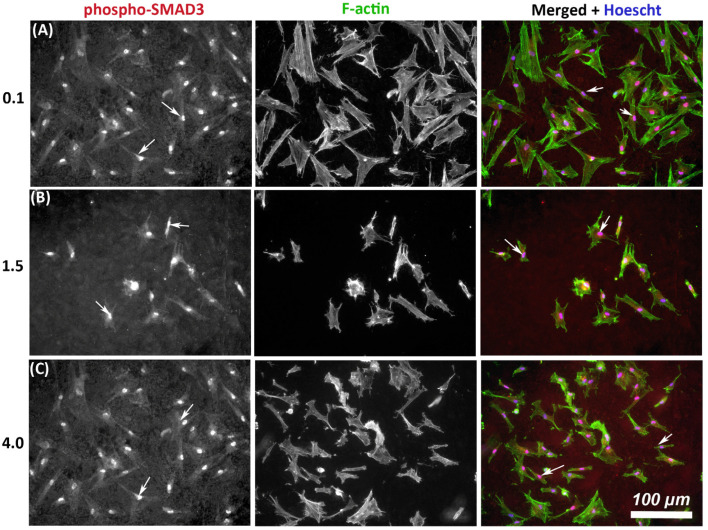
Influence of titanium substratum roughness on nuclear translocation of phospho-SMAD3 in HGFs. Cells were seeded on each topography and cultured for 24 h before being labeled with antibodies specific for phospho-SMAD3 which mediates transcriptional effects of TGF-β stimulation. Representative images are shown from 3 of the topographies employed, but nuclear translocation of phospho-SMAD3 was evident in HGFs on all tested topographies. Cells were co-labeled with FITC-phalloidin for F-actin to show cell morphology. White arrows denote nuclear translocation of phospho-SMAD3. Images are shown of cells cultured on surfaces of R_a_ = (**A**) 0.1, (**B**) 1.5, and (**C**) 4.0 demonstrating similar levels of nuclear translocation irrespective of substratum roughness.

**Figure 5 materials-14-06447-f005:**
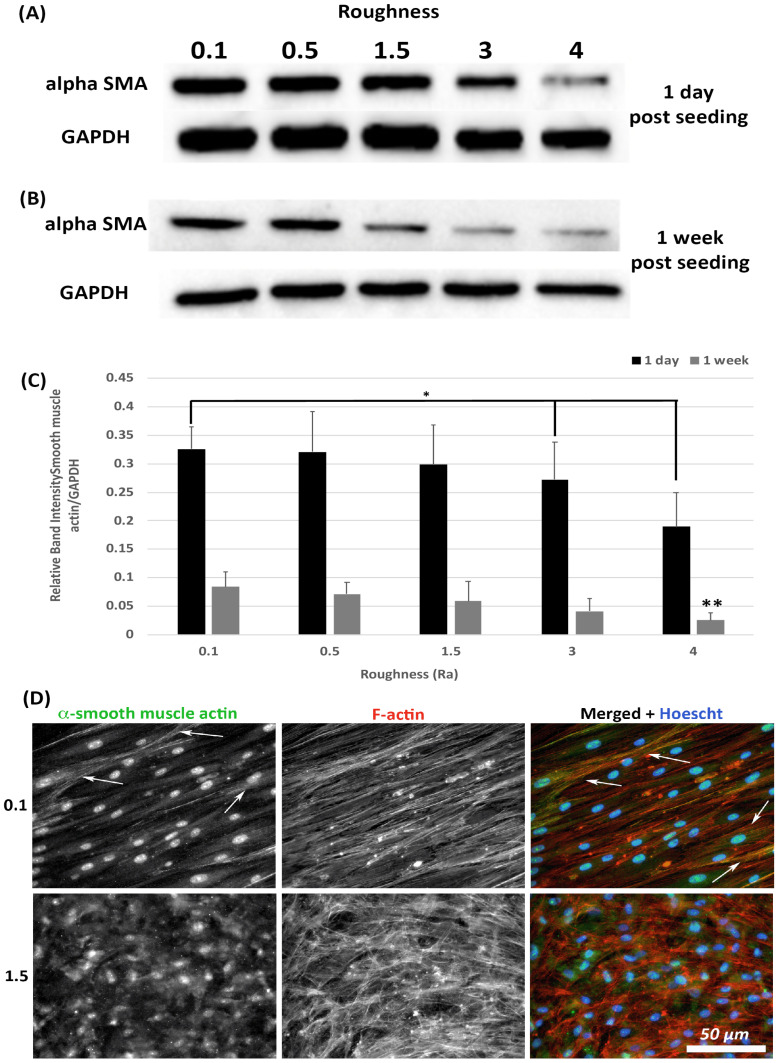
Expression of α-SMA in response to altered substratum roughness. HGFs were cultured on each topography and protein extracted at (**A**) 24-h and (**B**) 7-days post seeding to (**C**) quantify protein expression of α-SMA. At 24-h post seeding, surfaces with R_a_ = 3.0 and 4.0 significantly reduced α-SMA expression (* = *p* < 0.05 vs. cells grown on 0.1 at 1 day). At 7-days post seeding, only R_a_ = 4.0 significantly reduced α-SMA protein expression compared to PT (** = *p* < 0.05 vs. cells grown on 0.1 at 7 days). One-way ANOVA followed by a Bonferroni adjustment. All experiments were run in triplicate with cells isolated from 3 different individuals. (**D**) Cells were cultured on each topography for 7 days and labelled with antibodies specific to α-SMA and FITC phalloidin to visualize F-actin stress fiber networks. Representative images are shown from 0.1 and 1.5. Only HGFs cultured on R_a_ = 0.1 exhibited high levels of α-SMA incorporation into stress fiber bundles (indicated by white arrows).

**Figure 6 materials-14-06447-f006:**
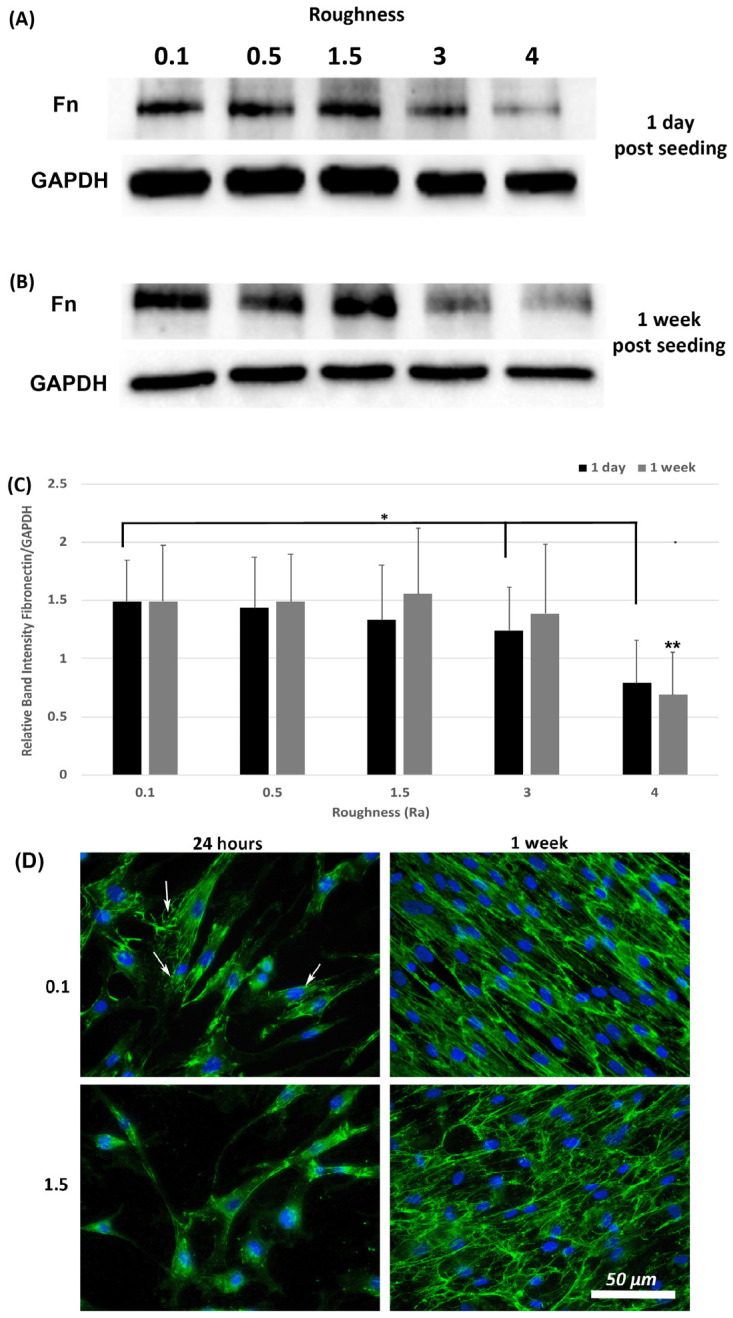
Expression of fibronectin in response to altered substratum roughness. HGFs were cultured on each topography and protein extracted at (**A**) 24-h and (**B**) 7-days post seeding to (**C**) quantify protein expression of fibronectin. At 24-h post seeding, surfaces with R_a_ = 3.0 and 4.0 significantly reduced fibronectin expression (* = *p* < 0.05 vs. cells grown on 0.1 at 1 day). At 7-days post seeding, only R_a_ = 4.0 significantly reduced fibronectin protein expression compared to 0.1 (** = *p* < 0.05 vs. cells grown on 0.1 at 7 days). One-way ANOVA followed by a Bonferroni adjustment. All experiments were run in triplicate with cells isolated from 3 different individuals. (**D**) Cells were cultured on each topography for 7-days and labeled with antibodies specific to fibronectin. Representative images are shown from 0.1, 1.5 and 4.0. Only HGFs cultured on R_a_ = 0.1 exhibited fibronectin fibril formation at 24-h post seeding (indicated by white arrows).

**Figure 7 materials-14-06447-f007:**
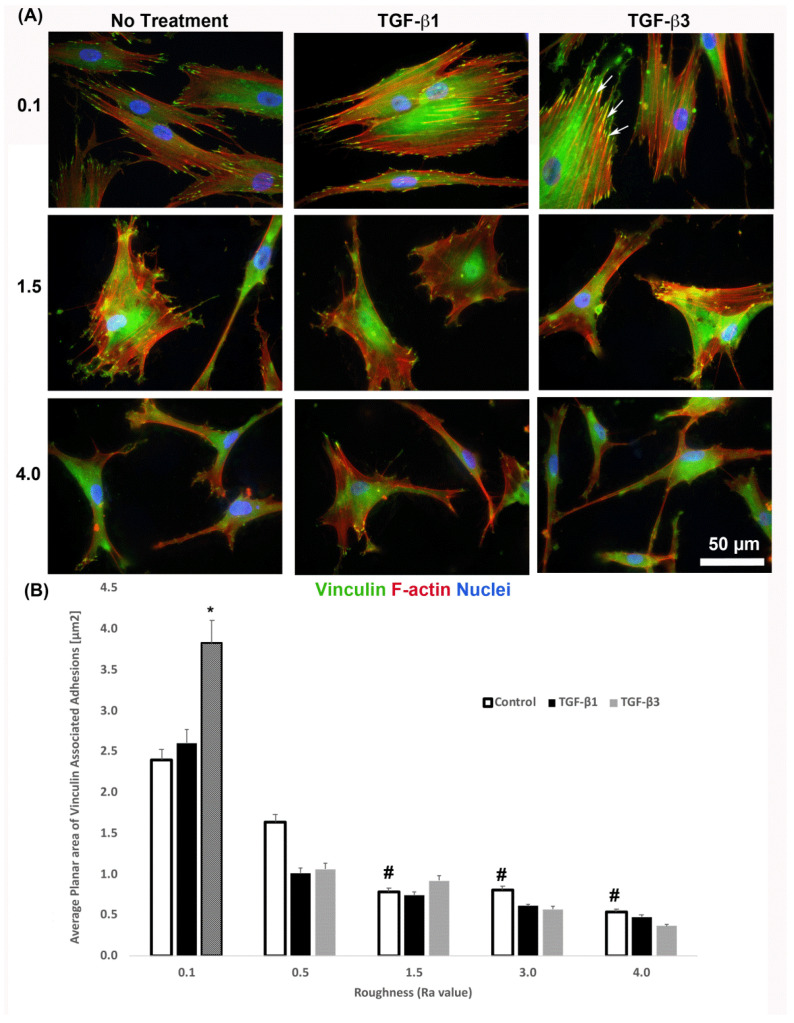
Influence of TGF-β isoforms on adhesion size in HGFs. (**A**) Immunocytochemistry for vinculin (green) and F-actin (red) in cells on 0.1, 1.5 and 4.0. Representative images are shown. TGF-β3 increased vinculin plaque size (denoted by white arrows). (**B**) Quantification of adhesion size in HGFs in response to TGF-β1 and TGF-β3 on 0.1 surfaces. Adhesion sizes were increased in the presence of TGF-β3 compared to untreated R_a_ = 0.1 (* *p* < 0.05 vs. cells grown on 0.1 at 1 day without TGF-β3). Adhesion size decreased on topographies with increased R_a_ (^#^ = *p* < 0.05 grown on 0.1 at 1 day). One-way ANOVA followed by a Bonferroni adjustment. All experiments were run in triplicate with cells from 3 different individuals.

**Figure 8 materials-14-06447-f008:**
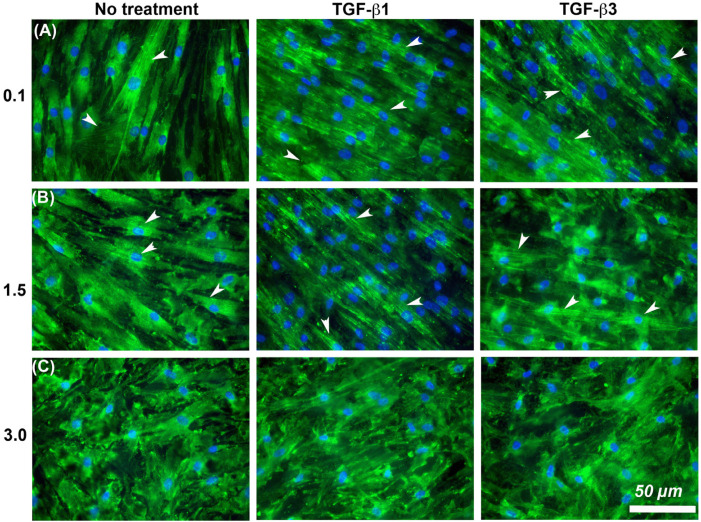
Influence of TGF-β isoforms and substratum roughness on α-SMA stress fiber formation in HGFs. (**A**) Stimulation of HGFs on R_a_ = 0.1 for 7 days with TGF-β1 and TGF-β3 increased α-SMA incorporation into stress fibers in comparison with untreated controls. (**B**) On surfaces of R_a_ = 1.5, both exogenous TGF-β1 and TGF-β3 increased myofibroblast differentiation versus controls, (**C**) but this was inhibited on R_a_ = 3.0. White arrows indicate fully differentiated myofibroblasts.

## Data Availability

All data supporting the results is available upon request.

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
