# Peer review of "Titanium Substratum Roughness as a Determinant of Human Gingival Fibroblast Fibronectin and α-Smooth Muscle Actin Expression"

_materials, 2021, doi:10.3390/ma14216447_

Round 1

Reviewer 1 Report

Li et al., shown how variations in titanium roughness (a material used for dental implants) impact the biology of gingival fibroblast by altering adhesion and the regulation of fibronectin and -SMA expression. Their findings highlight the importance of reducing adhesion size in the substratum to decrease profibrotic response around the implants. The authors manifest the necessity to further research on this topic. 

In general, the manuscript is interesting and well described, with good English.

However, I highlighted some issues:

  1. Line 63: Please correct “formation” to formation.
  2. Line 70: Ra definition should be described at line 65 instead of 70. 
  3. I suggest extent description of section 3.1. Adhesion formation. In particular the changes of cell morphology for Ti topographies. 
  4. I would place first section 3.2 before 3.1 for a better understanding.
  5. Respectfully, I can´t understand why the authors indicate that there are no differences in the cell adhesion among different roughness of Ti in figure 3A. Regarding, the authors start with 10000 cells for attachment assay. If well at the first hours there is no difference, it appears that there is a decreasing of 1.8-fold in Ra=4 respect 0.1. In other words, while Ra 0.1 shows an adherent capacity of 70%, Ra= 4 shows only 40% at 24 hr.
  6. Line 229: I think you mean alpha-SMA, please correct to α-SMA. Other mistakes can also be found here: lines 210, 211, 229.
  7. Can the authors maintain the colors indicated in the heading for every protein indicated in Figures 4 and 5D?
  8. Line 230: In the figure legend, the authors indicate “Representative images are shown from 0.1, 1.5 and 4”, but there is a lack the images for the last. A similar mistake is observed in figure legend 6, line 260.

Please double-check the order of the figures. On pages 12 and 13, it shown figures 7 and 8, but not 9 or 10.

Author Response

We thank the reviewers for their thoughtful and constructive comments on our manuscript. Based on these comments from the reviewers we have revised the manuscript. Reviewer 1 thought "the manuscript is interesting and well described", and reviewer 2 that it was "scientifically correct, it is well written, results are novel and within the scope of the journal". We respectfully think we have addressed the comments where possible based on the time constraints on resubmission. We address each individual comments as follows:

Reviewer 1:

1. Line 63: Please correct “formation” to formation.

We have made the correction in the manuscript as requested.

2. Line 70: Ra definition should be described at line 65 instead of 70.

We have moved the Ra definition to line 65 for defining it when it is first mentioned in the text.

3. I suggest extent description of section 3.1. Adhesion formation. In particular the changes of cell morphology for Ti topographies.

We have modified the section to better describe both the adhesions and cellular spreading as requested.

4. I would place first section 3.2 before 3.1 for a better understanding.

We have moved section 3.2 to 3.1. We agree with the reviewer that this order is more appropriate.

5. Respectfully, I can´t understand why the authors indicate that there are no differences in the cell adhesion among different roughness of Ti in figure 3A. Regarding, the authors start with 10000 cells for attachment assay. If well at the first hours there is no difference, it appears that there is a decreasing of 1.8-fold in Ra=4 respect 0.1. In other words, while Ra 0.1 shows an adherent capacity of 70%, Ra= 4 shows only 40% at 24 hr.

Although there are trends towards differences in adhesion, statistical analysis using ANOVA with a Bonferroni adjustment did not identify significant differences between the groups. Although the data trends in the direction of differences, there is a larger variation in effect size between patients, which is due to the inherent variability evident when using primary human cultures.

6. Line 229: I think you mean alpha-SMA, please correct to α-SMA. Other mistakes can also be found here: lines 210, 211, 229.

We have made the correction in the manuscript as requested.

7. Can the authors maintain the colors indicated in the heading for every protein indicated in Figures 4 and 5D?

This added to all figures as requested.

8. Line 230: In the figure legend, the authors indicate “Representative images are shown from 0.1, 1.5 and 4”, but there is a lack the images for the last. A similar mistake is observed in figure legend 6, line 260.

We have made the correction in the manuscript and added the images in to the figures.

9. Please double-check the order of the figures. On pages 12 and 13, it shown figures 7 and 8, but not 9 or 10.

This was an oversight on our proofreading of the manuscript and has been corrected.

Reviewer 2 Report

Manuscript entitled "Titanium Substratum Roughness as a Determinant of Human Gingival Fibroblast Fibronectin and α-Smooth Muscle Actin Expression" showed that the potential of human gingival fibroblasts to myofibroblastic transitions (FMT) was modulated by the properties of titanium substratum. FMT potential was determined by the levels of myofibroblast-related proteins: fibronectin and α-SMA. In this study, Authors showed also the intracellular localization of pSmad3 – TGF-β-activated transcription factor for pro-fibrotic genes, as well as the pattern of focal adhesion sites (FAs) in the HGF populations. Authors conclude that reduction of FAs size in HGFs by the appropriate properties of titanium substratum may be the useful tool for the reduction of pro-fibrotic response in these cells, which have a relevant clinical implication. The manuscript sounds scientifically correct, it is well written, results are novel and within the scope of the journal. However, there are a few issues that should be addressed before publication:

Comment 1: The Authors should complete the precise information on the cytokines used in this study: TGF-β1 and TGF-β3. The Materials and methods section should contain information about manufacturer or source of cytokines, used concentration and the time of HGFs exposition on TGF-β.

Comment 2: I suggest that Authors should complete the information about the antibody detecting phospho-SMAD3. It would be nice if the authors added information about the phosphorylation sites of anti-pSmad3 used antibody, for example: phosphoSmad2 (Ser465/467).

Comment 3: It is not clear what test was used to check the normal distribution before the using ANOVA statistical test for performed analyses.

Comment 4: It would strengthen the study if the size of focal adhesion sites were quantified and presented on graph in the figure 2.

Comment 5: Is the specific reason, why the Authors completely omitted the analyses of phosphorylation level and intracellular localization of phosphoSmad2? It is well known that pSmad2 similarly to pSmad3 play an important role in the activation of pro-fibrotic response in different cells. Can Authors explain this fact?  

Comment 6: It would strengthen the study if the quantification of Smad3 phosphorylation (measured by the fluorescence intensity of nuclear area in relation to DNA fluorescence) as well as western blot analyses of phosphorylation level of Smad2 and Smad3 were presented in the figure 4.

Comment 7: In the figure 5d Authors present the α-SMA staining in the HGFs cultured on the two different roughness of titanium substratum (1.0 and 1.5), whilst in the description is also 4.0. Can the Authors supplement this figure with the missing images?

Comment 8: Images in the figure 5d presenting α-SMA staining: can authors explain the presence of strong signal from α-SMA staining localized in the nuclear area? In properly performed staining should not be observed nuclear localization of cytoskeleton proteins.

Comment 9: It would strengthen the study if the fluorescence intensity of fibronectin staining were quantified and presented on graph. Authors should supplement this figure with the missing images presenting staining of fibronectin in HGFs cultured on the Ra=4.0 surfaces according to the figure description.

Comment 10: Can the authors explain why TGF-β1 and TGF-β3, but not TGF-β2 was used in this study?

Comment 11: The manuscript should be reread carefully and correct mistakes (stress fibres, not stressfibres – correction in whole manuscript; page 2 line 63: formation not formaiton; page 7 line 210, 211, page 8 line 229: lack of symbol α in α-SMA; page 14 line 367 – lack of symbol β).

Author Response

We thank the reviewers for their thoughtful and constructive comments on our manuscript. Based on these comments from the reviewers we have revised the manuscript. Reviewer 1 thought "the manuscript is interesting and well described", and reviewer 2 that it was "scientifically correct, it is well written, results are novel and within the scope of the journal". We respectfully think we have addressed the comments where possible based on the time constraints on resubmission. We address each individual comments as follows:

Reviewer 2:

  1. The Authors should complete the precise information on the cytokines used in this study: TGF-β1 and TGF-β3. The Materials and methods section should contain information about manufacturer or source of cytokines, used concentration and the time of HGFs exposition on TGF-β?

This information has been added to the manuscript. The growth factors were purchased from R&D systems and used at a concentration of 5 ng/ml as we have previously validated in other studies that they are effective at this concentration.

  1. I suggest that Authors should complete the information about the antibody detecting phospho-SMAD3. It would be nice if the authors added information about the phosphorylation sites of anti-pSmad3 used antibody, for example: phosphoSmad2 (Ser465/467).

This information has been added to the manuscript. Our Smad3 antibody is phosphorylated on Serine 423 and Serine 425

  1. It is not clear what test was used to check the normal distribution before the using ANOVA statistical test for performed analyses.

This information has been added to the manuscript. A normal probability plot was used and we also utilized Kurtosis score, the latter to assess large deviations.

  1. It would strengthen the study if the size of focal adhesion sites were quantified and presented on graph in the figure 2.

This information is already contained within the graph in figure 7b.

  1. Is the specific reason, why the Authors completely omitted the analyses of phosphorylation level and intracellular localization of phosphoSmad2? It is well known that pSmad2 similarly to pSmad3 play an important role in the activation of pro-fibrotic response in different cells. Can Authors explain this fact?  

We are aware that pSmad2 forms a complex with pSmad3 and pSmad4 prior to entering the nucleus to facilitate the effects of TGF-β  signaling. pSMAD3 is known to be induced in response to TGF-β signaling (J Dermatol Sci 2021 Sep 3;S0923-1811(21)00203-6), including in the gingiva (J Dermatol Sci, 2009 Dec;56(3):168-80). Respectfully, as pSmad2 and 3 form a complex with pSmad4 in response to TGF-β, we do not think analyzing pSmad2 would provide any further insights compared to the data we include on pSmad3.

  1. It would strengthen the study if the quantification of Smad3 phosphorylation (measured by the fluorescence intensity of nuclear area in relation to DNA fluorescence) as well as western blot analyses of phosphorylation level of Smad2 and Smad3 were presented in the figure 4.

We agree with hindsight that this would potentially strengthen the manuscript. However, we initially were going to have this data as supplementary information as it was simply done to confirm whether TGF-β signaling was active in HGFs in response to the different levels of titanium roughness. To this end, the ICC data answered our question by demonstrating that TGF-β  signaling is active in cells cultured on all surfaces. We are currently investigating in more depth the downstream mediators associated with TGF-β signaling and how this differs on different levels of roughness. We have identified 14 TGF-β signaling linked genes that show differential regulation particularly on surfaces of Ra = 3.0 and 4.0 Vs 0.1. Respectfully, this will however be a separate manuscript.

  1. In the figure 5d Authors present the α-SMA staining in the HGFs cultured on the two different roughness of titanium substratum (1.0 and 1.5), whilst in the description is also 4.0. Can the Authors supplement this figure with the missing images?

We have added the additional images for 4.0 and apologise for this oversight in the original manuscript.

  1. Images in the figure 5d presenting α-SMA staining: can authors explain the presence of strong signal from α-SMA staining localized in the nuclear area? In properly performed staining should not be observed nuclear localization of cytoskeleton proteins.

α-SMA is known to co-localize to invaginations of the nuclear membrane although is not present in the nucleoplasm itself (Histochem Cell Biol 2007 May;127(5):523-30). Indeed, when fibroblasts stretch, it is common for α-SMA to localize to the nuclear area (Histochem Cell Biol 2006 May;125(5):487-95).  The general findings and potential role of actin isoforms in the nucleus has also been the subject of high-profile reviews (Nat Rev Mol Cell Biol 2004 May;5(5):410-5; FEBS Lett 2008 Jun 18;582(14):2033-40.). As such, the localization of α-SMA seen in our study is not unexpected, but the majority of the staining is associated with the cytoplasm and stress fibres.

  1. It would strengthen the study if the fluorescence intensity of fibronectin staining were quantified and presented on graph. Authors should supplement this figure with the missing images presenting staining of fibronectin in HGFs cultured on the Ra=4.0 surfaces according to the figure description.

We respectfully feel that addition of fluorescence quantification would not increase the strength of the study. We have used the exact same conditions and timepoints and quantified the fibronectin levels using Western blotting, which we feel is much more appropriate

  1. Can the authors explain why TGF-β1 and TGF-β3, but not TGF-β2 was used in this study?

We selected TGF-β1 and TGF-β3 based on their roles in wound healing and gingival repair in particular. As stated in the manuscript, TGF-β1 is considered to be pro-fibrotic and some evidence suggests that TGF-β3 is anti-fibrotic, although the latter remains somewhat controversial with conflicting results in the literature. This was the primary reason we used these two isoforms, with TGF-β2 considered similar to TGF-β1 although known to be 100-1000 fold less potent in vitro with respect to cell response (for example, see Biochemistry 2014, 53, 36, 5737–5749). In general, in wounds were scarring is present, high levels of TGFβ1 and TGFβ2, and low levels of TGFβ3 are present, while in scar-free healing as associated with gingival tissue and fetal healing, high levels of TGFβ3 and low levels of TGFβ1 and TGFβ2 are present (J Dev Biol 2016 Jun 22;4(2):21).

  1. The manuscript should be reread carefully and correct mistakes (stress fibres, not stressfibres – correction in whole manuscript; page 2 line 63: formation not formaiton; page 7 line 210, 211, page 8 line 229: lack of symbol α in α-SMA; page 14 line 367 – lack of symbol β).

We apologise for these errors and have carefully proofread the revised manuscript.

Reviewer 3 Report

  1. Page 2, line 86. As controls, pickled titanium (PT, Ra = 0.1) and SLA topographies (Ra = 4.0) were kindly supplied by Institut Straumann AG.” It is not clear if you have used both the PT and SLA (Ra=4.0) as controls or as experimental groups for your experiments. Furthermore, there is no indication regarding the behavior of HGF on the Cell Tissue Plastic (TCP). Please clearly indicate the experimental and control groups that you have used.
  2. Page 3. Fig.1 It will be more aesthetic if all pictures present the same contrast and luminosity
  3. Page 3, line 109. For experimental reproducibility of the data, it will better if the cells were between the 3rd and the 5th passage.
  4. Page 5, line 159. It will be better if you clarify in the materials and methods section if you have used each primary culture, derived from the biopsy taken from each patient to perform your triplicates in the independent runs of your experiments. Why did you not pool the cells?
  5. Page 6, Figure 3. I would suggest reporting the proliferation rate of the cells on the TCP at the same endpoints to verify the proliferative potential of primary HGF that you have isolated. Additionally, in Figure 3 (B) at 7 days error bars are extremely high. Could you please justify why?
  6. Page 8, line 228, Page 10, line 258, Page 12, line 295-296, could you please rephrase the sentence “All experiments were run in triplicate with cells from 3 cells from 3 different individuals.”

Page 8 lines 225-226,” At 7 days post-seeding, only Ra = 4.0 significantly reduced a-SMA protein expression compared to PT (**p<0.05)” From the Western blot films images results that a-SMA protein levels are reduced after 7 days on Ra=3 and Ra=4.  Furthermore, it is not clear which is the PT surface, please use the same names in the pictures and the text.

  1. Page 12, Figure 7. Instead of “control”, it will be better to use the term no treatment as reported in figure 8.
  2. Page 13, line 325 “Understanding how titanium of different Ra levels of influences HGF “please delete the second of

10 Page 16, line 445 Could you please eliminate the duplicates of the numbering in the bibliography list.

Author Response

We thank the reviewers for their thoughtful and constructive comments on our manuscript. Based on these comments from the reviewers we have revised the manuscript. Reviewer 1 thought "the manuscript is interesting and well described", and reviewer 2 that it was "scientifically correct, it is well written, results are novel and within the scope of the journal". We respectfully think we have addressed the comments where possible based on the time constraints on resubmission. We address each individual comments as follows:

Reviewer 3:

  1. Page 2, line 86. As controls, pickled titanium (PT, Ra = 0.1) and SLA topographies (Ra = 4.0) were kindly supplied by Institut Straumann AG.” It is not clear if you have used both the PT and SLA (Ra=4.0) as controls or as experimental groups for your experiments. Furthermore, there is no indication regarding the behavior of HGF on the Cell Tissue Plastic (TCP). Please clearly indicate the experimental and control groups that you have used.

PT (0.1) is used as a control as we have done in our other studies (Miron et al, Biomaterials, 2010; Kim et al, J Cell Mol Med, 2015). In this manuscript, we have used Ra = 4.0 (SLA) as a positive control for reduced fibrotic readouts based on our previous study (Kim et al, J Cell Mol Med, 2015). This has been made clear in the manuscript. With regards to the behaviour of the cells on TCP, we do not find this a relevant comparison to make. It is conceivably a more foreign surface for cell culture than titanium is, and ultimately the goal of this study was to compare cell response to different topographical features, not changes in material chemistry. 

2. Page 3. Fig.1 It will be more aesthetic if all pictures present the same contrast and luminosity

We have altered the brightness of the images as requested by the reviewer.

3. Page 3, line 109. For experimental reproducibility of the data, it will better if the cells were between the 3rd and the 5th passage.

In our previous experience, we have found that the cells are phenotypically stable up to passage 7. Beyond passage 10, the cells begin to flatten out and lose their characteristic elongated fibroblast morphology, which is concomitant with a reduced proliferative capacity. We have published numerous papers using gingival fibroblasts (Sci Rep. 2019 Feb 25;9(1):2708; J Cell Mol Med. 2015 Jun;19(6):1183-96; J Dent Res. 2013 Nov;92(11):1022-8; J Dent Res. 2010 Dec;89(12):1450-4) and are confident that the cells up to passage 7 are still representative of the overall cell phenotype.

4. Page 5, line 159. It will be better if you clarify in the materials and methods section if you have used each primary culture, derived from the biopsy taken from each patient to perform your triplicates in the independent runs of your experiments. Why did you not pool the cells?

We have rewritten the sentence to make clear what was done within and between each experimental run. Cells were isolated and used from 3 separate individuals. Each condition was run in triplicate with cells from each patient (n=1). Each experiment was run 3 times.

5. Page 6, Figure 3. I would suggest reporting the proliferation rate of the cells on the TCP at the same endpoints to verify the proliferative potential of primary HGF that you have isolated. Additionally, in Figure 3 (B) at 7 days error bars are extremely high. Could you please justify why?

The error bars at day 7 are high due to one sample exhibiting higher cell number than the other patient samples, but the trend is the same for all patients irrespective of the effect size. We left this data in as respectfully we feel that it is still valid and differences between individuals is something that should be highlighted rather than removed from any study in our opinion. We have commonly noted differences in effect size (While trend is similar) depending on the individual from which the cells have been isolated and indeed we have previously presented the data from each patient as individual graphs rather than grouping the data in one graph (J Dent Res. 2010 Dec;89(12):1450-4).

6. Page 8, line 228, Page 10, line 258, Page 12, line 295-296, could you please rephrase the sentence “All experiments were run in triplicate with cells from 3 cells from 3 different individuals.”

We have modified the sentence as requested.

7. Page 8 lines 225-226,” At 7 days post-seeding, only Ra = 4.0 significantly reduced a-SMA protein expression compared to PT (**p<0.05)” From the Western blot films images results that a-SMA protein levels are reduced after 7 days on Ra=3 and Ra=4.  Furthermore, it is not clear which is the PT surface, please use the same names in the pictures and the text.

Although the band is reduced, based on densitometric analysis, it was not significant on 3.0 Vs 0.1 at day 7. We have also removed the term PT and will refer to it as Ra = 0.1.

8. Page 12, Figure 7. Instead of “control”, it will be better to use the term no treatment as reported in figure 8.

We have modified the sentence as requested.

9. Page 13, line 325 “Understanding how titanium of different Ra levels of influences HGF “please delete the second of

We have modified the sentence as requested.

10. Page 16, line 445 Could you please eliminate the duplicates of the numbering in the bibliography list.

We have eliminated the duplicates in the numbering and we apologise for this error.